# Shop-R1: Rewarding LLMs to Simulate Human Behavior in Online Shopping via Reinforcement Learning

**Yimeng Zhang**[1,2]     **Tian Wang**[2]     **Jiri Gesi**[2]     **Ziyi Wang**[3]     **Yuxuan Lu**[3]

**Jiacheng Lin**[4]     **Sinong Zhan**[5]     **Vianne Gao**[2]     **Ruochen Jiao**[2]     **Junze Liu**[2]

**Kun Qian**[2]     **Yuxin Tang**[2]     **Ran Xue**[2]     **Houyu Zhang**[2]     **Qingjun Cui**[2]

**Yufan Guo**[2]                                                    **Dakuo Wang**[3]

[1]Michigan State University          [2]Amazon          [3]Northeastern University

[4]University of Illinois Urbana-Champaign          [5]Northwestern University

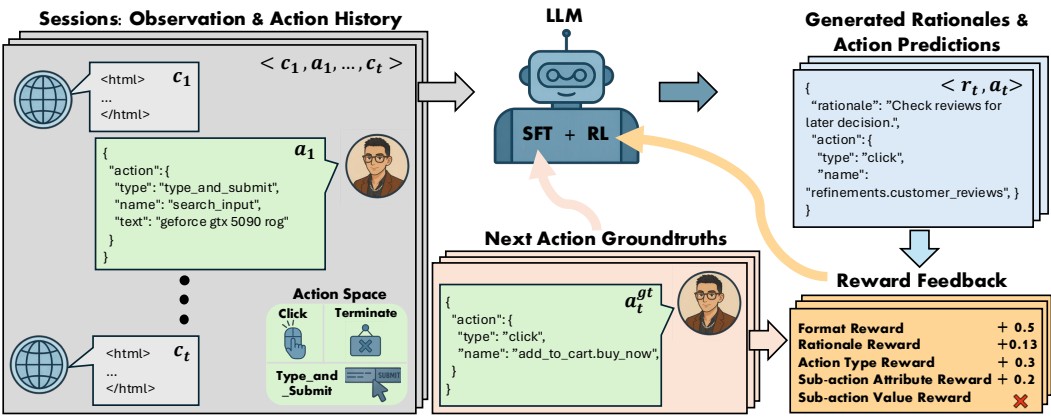

Figure 1: Overview of the proposed reinforcement learning framework, *Shop-R1*, designed to simulate real human behaviors in web-based shopping environments. Given an action history $a_{1...t-1}$ with corresponding website observations $c_{1...t-1}$, the model predicts the next action $a_t$ and its rationale $r_t$ based on the history and the latest website observation $c_t$. The generated responses are evaluated from four perspectives: format correctness, self-certainty of the rationale, action type accuracy, and sub-action (attribute and value) accuracy.

## Abstract

Large Language Models (LLMs) have recently demonstrated strong potential in generating 'believable human-like' behavior in web environments. Prior work has explored augmenting training data with LLM-synthesized rationales and applying supervised fine-tuning (SFT) to enhance reasoning ability, which in turn can improve downstream action prediction. However, the performance of such approaches remains inherently bounded by the reasoning capabilities of the model used to generate the rationales. In this paper, we introduce **Shop-R1**, a novel reinforcement learning (RL) framework aimed at enhancing the reasoning ability of LLMs for simulation of real human behavior in online shopping environments. Specifically, Shop-R1 decomposes the human behavior simulation task into two stages: rationale generation and action prediction, each guided by distinct reward signals. For rationale generation, we leverage internal model signals (e.g., logit distributions) to guide the reasoning process in a self-supervised manner. For

action prediction, we propose a hierarchical reward structure with difficulty-aware scaling to prevent reward hacking and enable fine-grained reward assignment. This design evaluates both high-level action types and the correctness of fine-grained sub-action details (attributes and values), rewarding outputs proportionally to their difficulty. Experimental results show that our method achieves a relative improvement of over 65% compared to the baseline. The project page is available at https://damon-demon.github.io/shop-r1.html.

# 1    INTRODUCTION

Large Language Models (LLMs) have shown remarkable performance in planning, reasoning, and decision-making tasks (Yao et al., 2023; Jin et al., 2025; Huang et al., 2024; Xu et al., 2025; Zhang et al., 2024b; Sun et al., 2025; Li et al., 2024; Gu et al., 2024; Jia et al., 2024; Zhang et al., 2024c; Chen et al., 2025b; Kong et al., 2025; Zhang et al., 2025). Recently, researchers have begun leveraging LLMs to simulate human behaviors in web-based environments, aiming to generate realistic, user-like action sequences on digital services (Chen et al., 2025a; Lu et al., 2025e; Wang et al., 2025a). This capability has promising applications across domains such as e-commerce (Kasuga & Yonetani, 2024; Khatuya et al., 2025), education (Yao et al., 2021), and social computing (Pan et al., 2006). Despite these advances, current LLM agents often fall short in producing behaviors that align with real humans. The most straightforward baseline is zero-shot prompting (Kong et al., 2023), where models are given textual instructions to imitate certain user types and output action sequences in a predefined format. While simple to implement, this method lacks the personalization and adaptability needed for high-fidelity behavior modeling (Lu et al., 2025a). To improve behavioral accuracy and reasoning coherence, recent work such as (Lu et al., 2025a) has introduced synthetic training data augmentation. Specifically, they use Claude 3.5 Sonnet (Anthropic, 2024) to generate rationales to create ⟨context, action, rationale⟩ triplets. These triplets are then used to perform supervised fine-tuning (SFT), enabling the model to learn both the actions and their underlying rationales. However, this approach faces the key limitations: the quality and diversity of rationales are ultimately constrained by the LLM used during data generation.

Since RL offers a flexible and effective training paradigm, particularly suited for settings with sparse and delayed feedback, and allows for fine-grained control over behavioral outputs (Kaufmann et al., 2024; Rafailov et al., 2023; Mu et al., 2024; Bai et al., 2022; Glaese et al., 2022), we utilize RL for simulating human shopping behavior, in contrast to prior work that focuses primarily on task completion (Lin et al., 2022; Zhou et al., 2023; Dong et al., 2025), where the user goal is clearly specified. In this work, we propose *Shop-R1*, a novel RL framework with hierarchical reward scheme designed to enhance LLMs for simulation of human online shopping behaviors. As shown in **Fig. 1**, Shop-R1 decomposes the human behavior simulation task into two stages: (1) *rationale generation* and (2) *action prediction*, with tailored reward signals for each component, since reasoning before action prediction has been shown to improve performance (Zhang et al., 2024a). For the reward design, we begin by introducing a binary format reward that encourages the model to produce responses in a parse-friendly structure, thereby facilitating reliable downstream evaluation and reward computation. Specifically, the model receives a non-zero reward only when its output conforms to the expected format; otherwise, it is penalized with zero reward. A more detailed discussion on the choice between zero and negative rewards is provided in **Appx. A**. For rationale generation, acquiring ground-truth rationales is inherently difficult. Although efforts like OPeRA (Wang et al., 2025c) attempt to collect self-reported rationales from real users, such annotations may omit implicit or unconscious decision factors. To address this, we incorporate a *self-certainty* reward (Kang et al., 2025; Zhao et al., 2025), quantified via the average Kullback–Leibler (KL) divergence (Csiszár, 1975) between the model's output distribution and a uniform distribution. This signal captures the model's confidence in its generated rationales, providing a supervision-free alternative to ground-truth rationales that contributes to training stability. For action prediction, we go beyond binary reward signals by introducing a *hierarchical reward scheme* that accounts for both action type and sub-action correctness. This design allows the agent to receive partial credit for plausible but imperfect behaviors, promoting smoother and more robust learning. Furthermore, to mitigate reward hacking and reflect the varying difficulty of different actions, we apply a difficulty-aware reward scaling strategy that adjusts the reward magnitude based on action complexity. Our **main contributions** are summarized as follows:

- To the best of our knowledge, we are the first to introduce RL into a simulation-oriented human behavior modeling task in web-shopping environments. We reformulate human online shopping behavior simulation as a two-stage prediction problem, comprising rationale generation and action prediction, and design distinct RL objectives for each.

- We introduce *Shop-R1*, a reinforcement-learning framework with a hybrid reward design. It integrates a *self-certainty* signal for rationale generation with a *hierarchical reward scheme* for action prediction. To ensure stable learning and prevent reward hacking, we further introduce a *format reward* and a *difficulty-aware reward scaling* mechanism.

- Experiments show that our proposed training pipeline achieves an exact match accuracy of 27.72%, outperforming supervised fine-tuning (16.76%) by over 65%, demonstrating the strong effectiveness of our approach in simulation-oriented human shopping behavior modeling. We further conduct a comprehensive ablation study to evaluate the contribution of each component in our design.

## 2 RELATED WORK

**LLM for human behavior simulation.** Large Language Models (LLMs) have emerged as powerful tools to simulate human behaviors in diverse real-world settings. Recent advances have led to the development of agent systems capable of generating plausible user actions based on static personas and interaction histories, enabling the modeling of behavior in contexts such as social science (Park et al., 2023a; 2024), recommender systems (Wang et al., 2023), e-commerce (Wang et al., 2025d), and user experience research (Lu et al., 2025d). These systems typically condition on user profiles (e.g., preferences, demographics) and session histories (e.g., clickstreams, task sequences) to predict the next likely user action, allowing for personalized and context-aware simulations. Beyond behavior prediction, recent efforts have enriched these simulations by incorporating explicit reasoning processes. Methods like ReAct (Yao et al., 2023) and reflection-based models (Shinn et al., 2023; Park et al., 2023b) prompt LLMs to produce intermediate thought traces before action generation, enhancing interpretability and decision quality. Systems such as WebAgent (Gur et al., 2023) and UX-Agent (Lu et al., 2025d) further decompose tasks into sub-goals using dedicated reasoning models, yielding improved control in complex environments like web interfaces. A parallel line of research explores agent-based LLM frameworks that simulate multi-agent interactions in dynamic environments (Ma et al., 2024; Wang et al., 2025b; OpenAI, 2025). These systems often adopt modular roles (e.g., planners, executors) and collaborative reasoning (Qian et al., 2024; Luo et al., 2024), offering insights into emergent social behaviors and teamwork dynamics. Despite recent advances, there remains a significant gap in exploring how RL can be leveraged to further enhance the simulation of human behavior using LLMs, particularly in the context of web-based shopping environments.

**Reward design for RL.** Reward design plays a central role in the effectiveness and generalization of RL algorithms, particularly in the context of aligning LLMs with desired behaviors. The prominent paradigm is Reinforcement Learning from Human Feedback (RLHF), which has been widely adopted to fine-tune LLMs using reward models trained on human preference data (Ouyang et al., 2022). While RLHF has demonstrated strong alignment capabilities, it is often bottlenecked by the high cost and limited scalability of collecting reliable human annotations (Touvron et al., 2023). Moreover, reward models themselves can introduce alignment biases and inaccuracies, especially when trained on limited or noisy preference comparisons (Gao et al., 2023). To alleviate these limitations, Direct Preference Optimization (DPO) (Rafailov et al., 2023) proposes a more efficient alternative that directly optimizes model parameters against human preference signals without an explicit reward model. Though computationally lighter, DPO and its variants still depend on the availability and quality of human-generated or approximated preference data, which can be inconsistent across tasks and domains. A complementary direction has emerged through Reinforcement Learning with Verifiable Rewards (RLVR), particularly suited for domains with deterministic correctness criteria such as code generation and mathematical reasoning (Guo et al., 2025; Su et al., 2025). RLVR frameworks employ rule-based verifiers to automatically compute reward signals based on strict correctness (e.g., exact string matching or functional equivalence) bypassing the need for human feedback. This shift toward automated objective reward functions has enabled the training of highly

capable models such as DeepSeek-R1 (Guo et al., 2025) and inspired new policy optimization methods such as GRPO (Shao et al., 2024) and its recent extensions (Yu et al., 2025; Liu et al., 2025). Despite these advances, reward design remains a fundamental challenge in RL for human behaviors. RLHF offers flexibility for modeling subjective tasks, but often suffers from scalability and reliability issues (Alsagheer et al., 2025; Lee et al., 2023; Casper et al., 2023; Moskovitz et al., 2023). In contrast, RLVR provides high precision by relying on clearly defined evaluation criteria, but is limited to tasks where such criteria exist (Su et al., 2025; Mroueh, 2025; Wen et al., 2025; Lu et al., 2025c). To address the unique challenges of simulating human online shopping behavior, we propose a hybrid reward framework specifically tailored to this domain. For rationale generation, the framework leverages internal model signals to compensate for the absence of ground-truth rationales in natural settings. For action prediction, it adopts a hierarchical reward structure with difficulty-aware scaling, which mitigates reward hacking and alleviates the inefficiency caused by prolonged zero-reward feedback.

## 3 METHODOLOGY

In this section, we first formulate the problem of human behavior simulation in the context of web-based shopping. We then present the design of our proposed RL framework, *Shop-R1*, tailored specifically for simulating human behavior in this setting. Although user persona and explicit intention can influence human behavior, such information is neither collected nor observable in real-world e-commerce logs, nor can it be reliably reconstructed from HTML content or clickstream sequences. Crucially, the objective of our task is not to infer latent psychological states, but to model how real users behave given the same observable environment. Since human behavior is inherently non-deterministic, the goal is not to predict a single "correct" action; rather, it is to learn the **distributional tendencies** that characterize real user decision-making.

**Problem statement.** In the context of web shopping, a user session is composed of a sequence of multi-step actions $a_{1...t...N}$, typically initiated with a search query and concluded by either a product purchase or a termination action (e.g., closing the browser). Following the setup of (Lu et al., 2025b), the action space comprises three primary action types: '*type_and_submit*', '*click*', and '*terminate*'. More details about the action space can be found in **Appx. B**. Each action $a_t$ is paired with a corresponding rationale $r_t$, which captures the user's underlying motivation or rationale at that time step. The model also receives contextual information, i.e., the observation space, representing the current state of the web environment. This context is encoded as a simplified HTML structure, as introduced in (Lu et al., 2025e), which preserves essential layout and content elements while discarding non-informative components such as scripts and styles. The task of human online shopping behavior simulation is formally defined as learning a function $f$ that predicts the next rationale $r_t$ and action $a_t$, given the cumulative context $c_{1...t}$ and action history $a_{1...t-1}$:

$$f(c_{1...t}, a_{1...t-1}) = r_t, a_t, \tag{1}$$

where $f$ denotes the model trained to simulate user behavior by generating the next-step rationale $r_t$ and action $a_t$ conditioned on the past context $c_{1...t-1}$, the corresponding past actions $a_{1...t-1}$, and the current context $c_t$. Thus, the model must jointly (a) understand the dynamic structure of the current web page, (b) integrate long-horizon action history, and (c) reason about the next plausible human action. This combination leads to long-horizon sequences and high-entropy free-form text predictions, making the task substantially more challenging than typical WebArena-style task-completion settings (Zhou et al., 2023). During the supervised fine-tuning stage, the rationales in training data are generated using LLMs and serve as supervision signals for a cold start. Need to note that *no* LLM-generated rationales are used during the subsequent RL stage.

**Cold start with SFT.** Following the approach of (Guo et al., 2025), we initialize the behavior simulation model $f$ through supervised fine-tuning (SFT) on annotated trajectories, where each rationale is generated by Claude 3.5 Sonnet (Anthropic, 2024) via Amazon Bedrock, without leveraging any user profile information. This SFT phase acts as a cold start for subsequent RL, grounding the model in realistic rationale and action patterns. During this phase, the model is trained to jointly generate rationales and corresponding actions. The training objective is to maximize the likelihood of the ground truth rationale-action pairs, conditioned on the the input query $q_t = c_{1...t}, a_{1...t-1}, r_{1...t-1}$:

$$L_{\text{sft}} = -\sum_{t=1}^{N} \log p(r_t, a_t \mid q_t), \tag{2}$$

This supervised initialization plays a crucial role in helping the model internalize the structural dependencies among context, rationale, and action early in the training pipeline. By grounding the model in these patterns upfront, we significantly enhance both the stability and sample efficiency of subsequent RL stages. More importantly, it provides an explicit signal for what constitutes a high-quality long-text output, such as correctly naming a clicked button or specifying a meaningful search query. These capabilities that are otherwise difficult to acquire solely through RL, especially given the sparse and delayed reward structure.

**Shop-R1.** To better guide policy optimization in the human behavior simulation setting, we decompose each step into two sub-tasks: rationale generation and action prediction. Each sub-task is assigned a tailored reward to improve alignment and interpretability. To ensure the ease and correctness of parsing predicted rationales and actions from model outputs, we introduce a *binary format reward*, which encourages the model to produce responses in a structured JSON format. This format adheres to a dictionary schema with two keys: *rationale* and *action*. For rationale generation, we employ a *self-certainty score* (Kang et al., 2025; Zhao et al., 2025), which quantifies the model's confidence in its generated rationale. Specifically, we compute the KL divergence between the model's predictive distribution over the vocabulary and a uniform distribution, averaged over the entire output sequence:

$$s(r_t \mid q_t) = \frac{1}{N|V|} \sum_{j=1}^{N} \sum_{i=1}^{|V|} p_{ij} \log \left( \frac{p_{ij}}{U_i} \right), \tag{3}$$

where $N$ is the number of tokens in the generated rationale $r_t$, $p_{ij}$ is the predicted probability of token $i$ at position $j$, and $U_i = \frac{1}{|V|}$ is the uniform distribution over the vocabulary $V$. Higher values of $s(\cdot)$ indicate greater certainty and consistency in the model's reasoning.

Table 1: Hierarchical reward schedule with Difficulty-Aware Reward Scaling (DARS). A response earns a format reward of 0.5 if it is in a valid JSON format; otherwise, it gains no format reward. A valid response can further gain partial credit for (i) the correct action type, (ii) the presence of the required sub-action attribute, and (iii) any long-text value prediction, whose reward equals the DARS factor multiplied by its ROUGE-L similarity to the ground truth.

| Action Type | Type Reward | Sub-action Attribute Reward | Text-Similarity Value Reward |
|---|---|---|---|
| *terminate* | 0.3 | None | None |
| *click* | 0.3 | +0.2 (if name $\neq \varnothing$) | +DARS × ROUGE-L(name) |
| *type_and_submit* | 0.3 | +0.1 (if name $\neq \varnothing$) +0.1 (if text $\neq \varnothing$) | +0.1 × ROUGE-L(name) +DARS × ROUGE-L(text) |

For action prediction, we replace the brittle binary signal with a *hierarchical reward scheme* that credits both the coarse-grained action type and its fine-grained sub-actions to stabilize training and discourage degenerate reward hacked policies. This hierarchical scheme densifies the reward landscape: it expands the set of profitable trajectories, lifts the agent out of the 'no-reward' plateau that typically stalls policy search, and makes reward hacking uneconomical. Concretely, every action, easy or hard, earns the *same* coarse-level reward once its high-level type is correct; only the more complex actions can unlock *additional* gains through their long-text sub-components. As a result, naively spamming the trivial '*terminate*' action no longer yields a competitive payoff, while executing the full ('*click*', '*type_and_submit*') sequence becomes the most lucrative strategy. Concretely, a '*click*' action containing a sub-action, specifying the button name to be clicked; partial rewards are granted for the correctly predicted components. Likewise, '*type_and_submit*' contains sub-action, providing the intended textual content. In contrast, '*terminate*' has no sub-actions and is scored only at the action-type level. Prediction accuracy is measured with task-specific metrics: discrete action types use an exact-match criterion, whereas free-form sub-actions are evaluated with ROUGE-L. A text-based sub-action, such as a button label or search query, earns a *soft* reward proportional to its ROUGE-L similarity to the ground truth, but only when that similarity exceeds a preset threshold (e.g., 0.75). Because long-text sub-actions are substantially harder, where modern webpages can expose thousands of candidate elements, we introduce a *difficulty-aware reward scaling* (DARS) factor that amplifies rewards for correctly predicting these components. This prevents reward hacking behaviors in which the agent repeatedly selects the trivial '*terminate*' action to secure easy points. The proposed hierarchical reward scheme is summarized in **Tab. 1**. Bringing these components together, the objective of *Shop-R1* is to maximize the combined reward signal derived from multiple

sources, while regularizing with a KL divergence to a reference policy:

$$\max_{\pi_\theta} \mathbb{E}_{r,a \sim \pi_\theta(q)} \left[ v(a) + \alpha s(r) + -\beta \, \text{KL} \left( \pi_\theta(r, a \mid q) \,\|\, \pi_{\text{ref}}(r, a \mid q) \right) \right], \quad (4)$$

where $\pi_{\text{ref}}$ denotes a fixed reference policy, $v(a_t)$ denotes the reward for action prediction and $\alpha$ and $\beta$ are hyperparameters that control the strength of the corresponding regularization terms. Need to note that because the action space used in our paper reflects general interaction patterns in Web GUIs, the proposed reward design can naturally generalize to different webpages and even to other simulation-oriented tasks involving Web GUI interactions. Importantly, the reward weights are not tuned for specific pages or environments. They function only as normalization constants to balance reward magnitudes, prevent trivial solutions (e.g., repeatedly predicting terminate), and ensure stable optimization rather than encoding task-specific biases.

## 4 EXPERIMENTS

### 4.1 EXPERIMENT SETUPS

**Datasets and models.** Our study is built on SHOP-CART dataset consisting of a corpus of 52,137 real-world shopping sessions collected from a leading global e–commerce service (Lu et al., 2025b). Each session logs the multi-turn interaction between a human customer and the website interface. More dataset details can be found in **Appx. C**. We enrich each recorded action with a natural language *rationale* automatically generated by Claude 3.5 Sonnet (see Appx. D for the prompting details). The provided observation context is formatted as simplified HTML (Lu et al., 2025e), which retains essential structural elements while filtering out irrelevant content such as scripts, styling information, and user-specific data. For SFT dataset, we keep each session intact. The model is asked to produce the assistant response, which contains both the rationales and the structured action prediction. For RL dataset, we convert a session into a sequence of <context, action> pairs. The context is the concatenation of (i) all previously observed contexts and (ii) the actions already taken; the target is the next action only. Because every session begins on the home page, there is always at least one observed <context, action> pair before the first prediction step, eliminating the open-world ambiguity of the very first move. To provide the model with slightly richer supervision on the harder behaviors, the two complex actions (*click* and *type_and_submit*) each occur about 10% more frequently than the simple *terminate* action. This mild skew prevents the learner from over-fitting to the trivial case while still maintaining near-uniform coverage, thereby supporting fair and informative per-class evaluation. All experiments fine-tune the publicly available `Qwen-2.5-3B-Instruct` model. The default 3B parameter backbone offers a favourable compute–performance trade-off.

**Baselines for comparison.** We evaluate our approach against several baseline schemes: (a) **Zero-shot prompting**, where the model generates outputs based solely on instruction prompts without additional training; (b) **RL (Binary)**, where the base model is optimized directly with RL, using only a sparse binary reward signal; (c) **SFT-only**, where the model is trained via supervised fine-tuning on data with LLM-generated rationales; (d) **SFT + RL (Binary)**, which extends SFT with reinforcement learning using a binary reward based on exact action match; and (e) **Shop-R1**, our proposed RL framework with hybrid reward design for the simulation-oriented human behavior modeling task.

**Training setups.** Our codebase is built on verl (Sheng et al., 2024), and all experiments were conducted on NVIDIA A100 GPUs (80 GB). We leveraged Fully Sharded Data Parallelism (FSDP) in PyTorch (Zhao et al., 2023) to maximize training efficiency. The default policy optimization algorithm is Group Relative Policy Optimization (GRPO) (Shao et al., 2024). Input sequences were padded or truncated to a maximum context length of 32k tokens, and the default sampling temperature is 0.6. We set the per-device batch size to 1, yielding a global batch size of 64. For supervised fine-tuning (SFT) we trained for 4 epochs with a learning rate of $2 \times 10^{-5}$; for reinforcement learning (RL) we trained for 500 steps with a learning rate of $1 \times 10^{-7}$. By default, we set the DARS factor to 1000, and use $\alpha = 0.005$ and $\beta = 0.001$ to weight the corresponding reward terms.

**Evaluation metrics.** We apply an exact match criterion for the accuracy evaluation of predicted user actions. A prediction is deemed correct only when every relevant component exactly matches the ground truth. For instance, in the case of '*click*' actions, both the specific subtype (such as clicking on a filter, search area, or another UI element) and the selected target must align with the true label. Similarly, for '*type_and_submit*' actions, the model should reproduce the similar meaning of input

text. Additionally, We report accuracy and F1 on the coarse-grained *action type* alone. Comparing these scores with exact-match accuracy highlights whether residual errors stem from misclassifying the high-level action type or from mistakes in the fine-grained label (button name or query text).

## 4.2 EXPERIMENTAL RESULTS

**Performance comparison with baselines.** Main performance comparison results are shown in **Tab. 2**. Firstly, zero-shot prompting yields low performance: without any task-specific adaptation Qwen-2.5-3B-Instruct achieves only 0.32% exact-action accuracy, confirming that long-horizon web behavior cannot be recovered from generic instruction tuning alone. Second, RL with sparse binary rewards on their own still fail to give the agent meaningful guidance. When we train the policy from scratch under this signal, it reaches only 1.01% exact-match action accuracy and 6.17% type accuracy. Third, a straightforward round of SFT is more effective, boosting performance to 16.76% exact match accuracy and 22.25% type accuracy. This confirms that dense, teacher-forced trajectories are crucial for injecting the structural knowledge (context → rationale → action) and illustrating the shape of the long-text fields (button labels or search queries) that the binary signal alone cannot convey. Fourth, appending an additional binary-reward RL phase after SFT delivers only mixed results: exact-match action accuracy actually slips to 16.55%, while type-level F1 rises to 28.07%. The agent thus learns to guess the coarse intent better, but it still struggles to reproduce the long-text values that drive the exact-match metric. In other words, the policy becomes better at guessing the coarse action type, but slightly worse at reproducing the fine-grained, long-text values required for an exact match.

Lacking finer-grained credit assignment, the binary objective cannot push the model beyond what SFT already achieves and in some respects even pulls it backwards. Training with this objective is furthermore prone to instability and converges substantially more slowly than optimization with richer, structured rewards. Our proposed *Shop-R1* framework closes most of this gap. By combining

Table 2: Simulation accuracy under different fine-tuning methods across models of different sizes. There are three complementary metrics: exact action accuracy (all sub-fields must match the label); action type accuracy, and action type F1 to disentangle mistakes in coarse intent classification from those in long-text arguments.

| Model | Settings | Exact Action | Action Type | |
|---|---|---|---|---|
| | | Acc. | Acc. | F1 |
| Qwen-2.5-3B-Instruct | Zero-shot prompting | 0.32% | 15.33% | 16.15% |
| | RL (Binary) | 1.01% | 6.17% | 9.92% |
| | SFT | 16.76% | 22.25% | 24.52% |
| | SFT + RL (Binary) | 16.55% | 23.74% | 28.07% |
| | *Shop-R1* (Ours) | **27.72%** | **36.40%** | **31.28%** |
| Qwen-2.5-1.5B-Instruct | Zero-shot prompting | 0.53% | 3.94% | 6.16% |
| | SFT | 10.86% | 23.58% | 29.02% |
| | *Shop-R1* (Ours) | 24.11% | 34.54% | 29.19% |
| Qwen-2.5-0.5B-Instruct | Zero-shot prompting | 6.76% | 12.88% | 15.55% |
| | SFT | 9.90% | 17.72% | 21.61% |
| | *Shop-R1* (Ours) | 27.72% | 31.83% | 21.20% |

hierarchical rewards, self-certainty signals, format rewards and difficulty-aware scaling, it delivers 27.72% exact-action accuracy (+65% relative to SFT) and pushes action-type accuracy and F1 to 36.40% and 31.28%, respectively. The simultaneous rise of both the coarse (type-level) and fine-grained (exact-match) metrics indicates that Shop-R1 not only identifies the correct intent more often, but also reproduces the long-text values (button labels, text queries) with higher fidelity. More performance comparison results on OPeRA dataset (Wang et al., 2025c) can be found in **Appx. E**.

As shown in **Tab. 3**, we decompose accuracy by action type. *Zero-shot prompting* shows the classic "intent–content" split. For example, it can guess that a '*click*' is needed (38.7% type accuracy) yet almost never names the exact UI target (0.58 % exact). Even if *SFT* can boost the performance but the gain remains uneven, which suggests that teacher forcing alone does not give the model enough credit assignment signal for predicting high-entropy arguments such as search queries. Appending a sparse binary RL phase after SFT still fails to boost these harder text-generation cases. *Shop-R1* reshapes those incentives, higher exact match accuracy is achieved, indicating that the model is no longer satisfied with merely selecting correct type but is learning to identify the correct widget and query text as well. To be summarized, dense and structured feedback is essential: it overcomes the no-reward plateau, and makes reward hacking uneconomical.

Table 3: Exact action accuracy and action type accuracy for each action type: '*click*', '*type_and_submit*', and '*terminate*', across different models and finetuning methods.

| Models | Settings | Exact Action Acc. Per Action Type | | | Action Type Acc. Per Action Type | | |
| --- | --- | --- | --- | --- | --- | --- | --- |
| | | *click* | *type_and_submit* | *terminate* | *click* | *type_and_submit* | *terminate* |
| Qwen-2.5-3B-Instruct | Zero-shot prompting | 0.58% | 0.15% | 0.00% | 38.7% | 1.62% | 0.00% |
| | SFT | 4.93% | 3.84% | 49.80% | 8.55% | 15.36% | 49.80% |
| | SFT + RL (Binary) | 8.12% | 3.25% | 45.51% | 17.25% | 13.88% | 45.51% |
| | *Shop-R1* (Ours) | 7.39% | 7.53% | 81.84% | 10.29% | 28.66% | 81.84% |
| Qwen-2.5-1.5B-Instruct | Zero-shot prompting | 1.01% | 0.15% | 0.39% | 10.00% | 0.44% | 0.39% |
| | SFT | 4.49% | 7.83% | 23.44% | 15.07% | 32.35% | 23.44% |
| | *Shop-R1* (Ours) | 3.62% | 8.12% | 72.85% | 6.52% | 34.12% | 72.85% |
| Qwen-2.5-0.5B-Instruct | Zero-shot prompting | 0.43% | 0.15% | 24.02% | 12.90% | 4.43% | 24.02% |
| | SFT | 3.19% | 7.68% | 21.88% | 5.94% | 26.59% | 21.88% |
| | *Shop-R1* (Ours) | 0.72% | 3.99% | 97.07% | 1.01% | 17.87% | 97.07% |

## 4.3 ABLATION STUDY AND ANALYSIS

**Model size.** **Tab. 2** and **Tab. 3** reveal a consistent scaling trend. In the zero-shot regime, the 3B backbone already outperforms its 1.5B and 0.5B counterparts by a factor of $\times 4 \sim 5$ on coarse action–type accuracy, confirming that larger models possess stronger out-of-the-box priors for human behavior simulation at the setting of website shopping. After SFT, all sizes gain, yet the improvement is more pronounced for the two smaller backbones than for the 3B model, suggesting that demonstration learning compensates for limited capacity. *Shop-R1* lifts every backbone to its best operating point, but the shape of the gains differs by scale. The 3B variant reaches the highest overall numbers while distributing its improvements evenly across the two complicated action types. By contrast, the 0.5B model achieves a comparable headline exact match accuracy (27.72%) almost entirely by over-predicting the easiest '*terminate*' action (97.07% exact) and largely ignoring the more semantically demanding classes. The 1.5B backbone sits in between, recovering moderate fidelity on '*click*' and '*type_and_submit*' while retaining a strong but not overwhelming bias toward '*terminate*'. In short, scaling primarily augments the model's ability to handle *long-text, high-entropy* actions; smaller networks can still match aggregate accuracy by exploiting the high-reward termination branch, but they do so at the cost of behavioral diversity. These findings underscore that, although Shop-R1 markedly mitigates capacity limitations, genuine mastery of simulation-oriented web-shopping action prediction tasks continues to benefit from larger backbones.

**Sampling temperature.** **Fig. 2** shows that *Shop-R1* is robust to sampling temperature, yet the three evaluation metrics react in distinct ways that reveal how temperature sampling propagates through the decision hierarchy. Action-type accuracy remains almost constant (36%) across the entire temperature sweep because this metric aggregates all predictions: small misclassifications in one direction are largely offset by fixes in another, leaving the overall hit rate unchanged. By contrast, the F1 score declines steadily (31.28% → 28.36%) as temperature rises; class-averaging penalizes any asymmetric increase in confusion. Interestingly, a modest boost from the default $\tau = 0.6$ to $\tau = 0.7$ improves exact-match accuracy to its peak of 28.63%: a trace of stochasticity helps the generated response escape local maxima and occasionally assemble the full long-text argument that greedy decoding would miss.

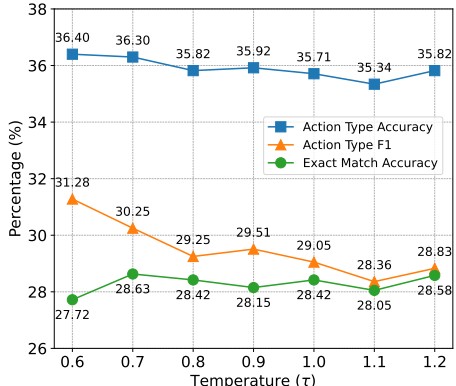

Figure 2: Sampling temperature ablation study.

When $\tau > 0.8$ the added entropy no longer uncovers new correct completions; instead it corrupts fine-grained fields faster than it fixes them, so exact-match plateaus while F1 continues to erode. This pattern is expected since the SFT stage already anchors the model to dataset-specific behavior, privileging faithful simulation over creativity. Taken together, these trends indicate that temperatures in the 0.6–0.8 band offer the best trade-off, preserving robust intent classification, maximizing strict exact-match, and avoiding the metric degradation that emerges once the sampler becomes overly exploratory.

**Training component.** **Tab. 4** makes clear that every element of *Shop-R1* addresses a different pathology. Removing the SFT warm-start cripples the agent: despite having all RL signals, exact-

Table 4: Ablation study on different training component configurations, evaluated by exact match action accuracy and action type accuracy / F1.

| Model | Training Scheme Components | | | | | Exact Action | Action Type | |
|---|---|---|---|---|---|---|---|---|
| | SFT | Format Reward | Rationale Reward | Reward Scale | Action Reward | Acc. | Acc. | F1 |
| Qwen-2.5-3B-Instruct | ✗ | ✓ | ✓ | ✓ | hierarchical | 4.63% | 36.56% | 21.92% |
| | ✓ | ✗ | ✓ | ✓ | hierarchical | 2.87% | 3.19% | 5.04% |
| | ✓ | ✓ | ✗ | ✓ | hierarchical | 26.93% | 37.25% | 33.74% |
| | ✓ | ✓ | ✓ | ✗ | hierarchical | 27.83% | 27.20% | 11.70% |
| | ✓ | ✓ | ✓ | ✓ | binary | 27.04% | 27.46% | 12.11% |
| | ✓ | ✓ | ✓ | ✓ | hierarchical | **27.72%** | **36.40%** | **31.28%** |

match drops to 4.63%, underscoring that a supervised prior is indispensable for learning the shape of long-text arguments. Omitting the format reward is even more destructive, where exact accuracy plunges to 2.87% and type-level metrics fall below 6% since unparseable JSON outputs earn zero credit, starving the learner of gradient signal. When the *self-certainty (rationale) reward* is ablated, coarse intent prediction remains strong but exact-match lags the full system by 0.8%, indicating that explicit feedback on the generated rationales mainly tightens the long-text portion of an action rather than its top-level label. Disabling the *difficulty-aware reward scaling* or reverting to a *binary* action reward leads to a different failure mode: the model still attains around 27% exact accuracy, yet type-level F1 degrades to 11–12%. Inspection shows that, without either scaling or hierarchical credit, the agent gravitates toward the easy high-reward '*terminate*' action and rarely ventures into harder '*click*' or '*type_and_submit*' cases, which is a classic reward-hacking pattern. The full configuration combines all signals and delivers the best balance demonstrating that *each component is necessary*: SFT injects linguistic priors, the format reward safeguards parsability, the self-certainty term refines long-text precision, and hierarchical difficulty-scaled rewards prevent degenerate policies while promoting fine-grained action fidelity.

**Whole-session v.s. latest-step context. Tab. 5** isolates the impact of including the *simplified HTML* of each visited page in the action history. Removing this structural cue slashes exact-match accuracy from 27.72% to 14.74%, a nearly 50% relative loss, while coarse action-type accuracy drops more modestly. The sharp divergence indicates that, although the model can still infer *which action type* of interaction is likely next from the dialogue trace alone, it struggles to generate the *fine-grained arguments*, the precise button label or query string, without access to the page's detailed context. Interestingly, the class-balanced F1 score rises slightly, suggesting that the only latest-step context variant compensates by spreading probability mass more evenly across action types, however, this redistribution does not translate into correct long-text completions. In short, supplying even a token-efficient, pruned HTML view is critical for high-fidelity simulation: it grounds the language model in the concrete UI affordances required for exact replay, and although it imposes a substantial overhead on the context window, this cost is justified by its necessity for accurate simulation.

Table 5: Comparison of model performance when using either the whole-session context or only the latest-step context as input.

| Context Settings | Exact Action | Action Type | |
|---|---|---|---|
| | Acc. | Acc. | F1 |
| whole-session | 27.72% | 36.40% | 31.28% |
| latest-step | 14.74% | 30.46% | 33.48% |

## 5 CONCLUSION

In this work, we introduced *Shop-R1*, a novel reinforcement learning framework tailored for simulating real human behavior in web-based environments using LLMs. By decomposing the task into two sub-problems, rationale generation and action prediction, and equipping each with carefully designed, structured reward signals, *Shop-R1* addresses key limitations of prior approaches relying solely on supervised fine-tuning or sparse binary rewards. Our hybrid reward scheme incorporating self-certainty scoring, hierarchical credit assignment, format regularization, and difficulty-aware scaling leads to substantial improvements in exact match accuracy and robustness across model sizes. Extensive experiments demonstrate that *Shop-R1* not only surpasses existing baselines by wide margins, but also mitigates common pathologies such as reward hacking and over-reliance on trivial actions. These findings highlight the promise of structured RL frameworks in enabling language agents to perform fine-grained, interpretable, and high-fidelity behavior simulation, paving the way for more realistic and personalized virtual user modeling in future interactive systems.

## LLM USAGE STATEMENT

Large Language Models (LLMs) were used solely as writing assistants. Specifically, the authors drafted the initial paragraphs of the paper, and then employed an LLM to polish the language for clarity and readability. The final paragraphs were obtained after multiple rounds of human–LLM interaction, with the authors carefully reviewing, editing, and approving all content. The research ideas, experimental design, implementation, and analysis were entirely conducted by the authors.

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

# APPENDIX

## A   THEORETICAL ANALYSIS: NEGATIVE REWARDS VS. ZERO REWARDS IN GRPO

A common concern in reinforcement learning is that shifting rewards by a constant value can introduce biases related to episode length (e.g., "survival" or "suicide" incentives) when optimizing for discounted cumulative returns ($G_t = \sum \gamma^k r_{t+k}$). However, in the context of Group Relative Policy Optimization (GRPO), we demonstrate that the optimization landscape is invariant to affine transformations of the reward function. Consequently, setting the failure penalty to zero or a negative value yields mathematically identical gradients.

### A.1   PROOF OF AFFINE INVARIANCE

Let $\mathcal{R} = \{r_1, r_2, \ldots, r_G\}$ be a set of rewards generated by a group of $G$ outputs for a single input query. GRPO computes the advantage $A_i$ for the $i$-th output via group-wise standardization:

$$A_i = \frac{r_i - \mu}{\sigma} \tag{A0}$$

where $\mu = \mathbb{E}[\mathcal{R}]$ is the group mean and $\sigma = \sqrt{\text{Var}(\mathcal{R})}$ is the group standard deviation.

Consider two reward schemes: *Scheme A* with rewards $r$ (e.g., failure penalty 0) and *Scheme B* with rewards $r'$ (e.g., failure penalty $n < 0$). We assume Scheme B maintains the relative order of Scheme A, meaning $r'$ is an affine transformation of $r$:

$$r' = \alpha r + \beta, \quad \text{where } \alpha > 0 \tag{A0}$$

For example, if Scheme A uses $\{1, 0\}$ for success/failure and Scheme B uses $\{1, -1\}$, then $\alpha = 2$ and $\beta = -1$. We now derive the advantage $A'_i$ for Scheme B.

First, the new mean $\mu'$ is:

$$\mu' = \mathbb{E}[\alpha r + \beta] = \alpha \mathbb{E}[r] + \beta = \alpha \mu + \beta \tag{A0}$$

Second, the new standard deviation $\sigma'$ is:

$$\sigma' = \sqrt{\text{Var}(\alpha r + \beta)} = \sqrt{\alpha^2 \text{Var}(r)} = |\alpha| \sigma = \alpha \sigma \tag{A0}$$

Substituting $\mu'$ and $\sigma'$ into the advantage formula:

$$A'_i = \frac{r'_i - \mu'}{\sigma'} = \frac{(\alpha r_i + \beta) - (\alpha \mu + \beta)}{\alpha \sigma} = \frac{\alpha (r_i - \mu)}{\alpha \sigma} = \frac{r_i - \mu}{\sigma} = A_i \tag{A0}$$

Since $A'_i = A_i$, the advantage values—and thus the policy gradients—are identical.

### A.2   NUMERICAL EXAMPLE

To illustrate this property with a realistic, unbalanced distribution, consider a group of $G = 4$ samples containing 1 success and 3 failures.

- **Scheme A (Zero Penalty):** Rewards are $\{1, 0, 0, 0\}$.
    - $\mu = 0.25$, $\sigma \approx 0.433$.
    - $A_{\text{success}} = \frac{1 - 0.25}{0.433} \approx \mathbf{1.73}$.
    - $A_{\text{failure}} = \frac{0 - 0.25}{0.433} \approx \mathbf{-0.57}$.
- **Scheme B (Negative Penalty):** Rewards are $\{1, -1, -1, -1\}$.
    - $\mu' = -0.5$, $\sigma' \approx 0.866$ (Note that $\sigma' = 2\sigma$).

$$- \; A'_{\text{success}} = \frac{1-(-0.5)}{0.866} \approx \mathbf{1.73}.$$
$$- \; A'_{\text{failure}} = \frac{-1-(-0.5)}{0.866} \approx \mathbf{-0.57}.$$

As shown, the computed advantages are identical. This confirms that within the GRPO framework, the choice between zero penalty and negative penalty does not affect the optimization trajectory, provided the relative ranking of rewards is preserved.

## B  SYSTEM PROMPT AND ACTION SPACE

```
<IMPORTANT>
Your task is to predict the next action and provide rationale for the
    action based on the previous actions and context.
You need to pretend that you are a user, browsing amazon.com and
    searching for a product to purchase.
The history action (with details described below) and context will be
    provided to you.
You need to predict the next action and provide rationale for the action.
</IMPORTANT>

# Action Space
An action is represented in JSON format, and there are three primary
    types of actions:
#### 1. `type_and_submit`:
Type text into an input field and immediately submit the form. Equivalent
     to typing text into an input and pressing enter key.
{
   "type": "type_and_submit",
   "name": "input_name",
   "text": "search_text"
}

#### 2. `click`:
Click on a button or clickable element identified by `name`.

{
   "type": "click",
   "name": "clickable_name"
}

#### 3. `terminate`:
When you are unsatisfied with the current search result and you don't
    want to buy anything, use `terminate` to indicate that you want to
    close the browser window and terminate the task.
{
   "type": "terminate"
}

# Context
Your context will be an **simplified version** of the raw HTML of the
    amazon page you are looking at. Some interactable elements will be
    added a unique "name" attribute, which you can use to identify the
    element to interact with (click or type_and_submit).

# Rationale
The rationale is a first-person sentence of what you are thinking when
    you make the action. It should be a short sentence that explains why
    you are making the action.

# Output Format
You need to predict the next action and provide rationale for the action.
     Your output should follow a strict JSON form:
{
   "rationale": "<rationale>", // rationale goes here, a string
```

```
    "action": {
        // action goes here
        "type": "<type>",
        ...
    },
}
<IMPORTANT>
OUTPUT A SINGLE JSON OBJECT, NOTHING ELSE.
</IMPORTANT>
```

As shown in the above system prompt, the human-behavior simulation task in web shopping is to predict the next rationale and action conditioned on (a) the entire past action history (past observations + past actions) and (b) the current web-page observation. The action space contains three types:'*type_and_submit*', '*click*', and '*terminate*'. For '*type_and_submit*', the model must generate an open-ended long-text argument (e.g., search queries), making this a high-entropy prediction task. For '*click*', the model must identify the correct clickable element name among hundreds to thousands of possible UI elements, which varies across pages and depends entirely on the current HTML observation.

## C  DATASET DETAILS

Our dataset is constructed from SHOP-CART dataset, which contains approximately 52k real-world shopping sessions collected from a global e-commerce platform (Lu et al., 2025a). The dataset spans a wide range of page layouts and user

| Dataset Split | click | type_and_submit | terminate |
|---|---|---|---|
| Train | 64,578 | 68,000 | 44,371 |
| Test | 690 | 677 | 512 |

Table A1: Action-type distribution of our large-scale web-shopping dataset.

interaction behaviors. Each action is formatted as a parseable JSON object (action type + sub-action fields) and paired with the corresponding web-page observation. The raw HTML is further simplified to remove non-essential content (e.g., scripts, CSS, visual-only elements) while preserving the interactive structure, enabling LLMs to process it more efficiently. From these real user sessions, we construct 176,949 training examples and 1,879 testing examples, where each example includes: (i) historical (observation, action) pairs, and (ii) the current simplified HTML observation. Basic action-type statistics are shown in Table A1.

## D  REASONING SYNTHESIZE PROMPT

```
You will be given a customer's shopping journey on one of the largest e-
    commerce services globally.
You will be given the context (what the user is looking at), the action (
    what the user did), and your job is to predict the user's rationale
    for the action.
The rationale should follow

Here is an example:
{example}

For each action in the input, output a rationale.

If the action is "terminate", it means that you didn't find any desired
    product and you decided to leave the website by closing the browser
    window.
```

## E  EXPERIMENTS ON OPERA DATASET

To verify the generality of our method, we have re-implemented Shop-R1 on the public OPeRA dataset (Wang et al., 2025c), which contains 692 real human web-shopping sessions, resulting in 8,212 training and 1,508 testing examples. The dataset statistics are summarized in Tab. A2. As shown in Tab. A3, the performance trends on OPeRA are consistent with those observed on our in-house dataset: (1) Zero-shot prompting performs poorly; (2) SFT substantially improves results; (3) Our proposed Shop-R1 (SFT + RL) yields the largest gain, particularly in exact-action accuracy, which is the most challenging metric. These findings confirm that Shop-R1 generalizes effectively to public datasets.

| Dataset Split | *input* | *click* | *scroll* |
|---------------|---------|---------|----------|
| Train | 499 | 4379 | 3334 |
| Test | 107 | 856 | 545 |

Table A2: Statistics of the OPeRA dataset used in our re-implementation.

| Model | Settings | Exact Action Acc. | Action Type Acc. | Action Type F1 |
|-------|----------|-------------------|------------------|----------------|
| **Qwen2.5-VL-3B-Instruct** | Zero-shot Prompt | 6.41% | 34.45% | 38.79% |
| | SFT | 20.23% | 60.86% | 53.95% |
| | SFT + RL | 38.44% | 57.27% | 57.69% |
| **Claude-3.5-Sonnet** | Zero-shot Prompt | 7.66% | 58.83% | 45.61% |

Table A3: Performance comparison on the public OPeRA dataset.

