# OpenReview forum: "Shop-R1: Rewarding LLMs to Simulate Human Behavior in Online Shopping via Reinforcement Learning"
_ICLR.cc/2026/Conference — ICLR 2026 Poster_

### Official Review · Reviewer_5Pw3 · 2025-10-28

**Soundness:** 1
**Presentation:** 2
**Contribution:** 1
**Rating:** 2
**Confidence:** 4

**Summary:**

A user simulation of online shopping, Shop-R1, is proposed in this work. The user model will generate actions with reasoning, and is improved by supervised learning and reinforcement learning. To prevent reward hacking and accurately represent the difficulty of specific actions, a hierarchical reward signal is proposed. However, the method is only tested with Qwen variants models and compared with limited baselines, which should be further improved in the future version.

**Strengths:**

* The authors conduct a detailed ablation study across different training configurations, e.g. different reward signals and temperature of sampling.

**Weaknesses:**

* Leveraging supervised learning and reinforcement learning for user simulation is a well-studied area. For example, the user model in GenTUS [1] is first trained with supervised learning and then further optimised through reinforcement learning. Similarly, USimAgent [2] simulates user clicking behaviours, including reasoning processes. Therefore, the novelty and contribution of this work should be clarified and elaborated further, which means the first contribution mentioned in the introduction could be an overstatement, where the authors claim they are the first to introduce RL into simulation-oriented human behaviour modelling.
* Building on the previous point, existing reinforcement learning and user simulation methods should be more carefully examined, as they may already address some of the challenges discussed in this paper. For instance, to prevent the model from selecting trivial actions (e.g., always choosing termination), a user goal and goal-failure penalty can be incorporated. Moreover, the choice of hyperparameters (e.g., the weight of the type reward) requires stronger justification. In addition, “accuracy” or “F1 score” may not be ideal metrics for evaluating user models, since multiple user behaviours can be equally reasonable. More suitable alternatives could include direct evaluation (e.g., human judgments of naturalness) or indirect evaluation (e.g., assessing the downstream system trained on simulated users).
* The model selection is also limited (only Qwen variants are tested), making it difficult to assess the generalizability of the proposed framework. Furthermore, the absence of proper baselines weakens the validity of the claimed contributions.

[1] Lin et al. GenTUS: Simulating User Behaviour and Language in Task-oriented Dialogues with Generative Transformers. SIGDIAL 2022.

[2] Zhang et al. USimAgent: Large Language Models for Simulating Search Users. SIGIR 2024.

**Questions:**

* In L081, the human behaviour is divided into two stages. Is there any theory or research supporting this setting? Is this really how humans act during web shopping?
* Typo: The "Sec. A" should be "Appendix A" in L177. In addition, the details of the action space are still unclear, since only a prompt is provided in the appendix.
* Why is there no negative reward in the proposed training framework?

---

> ### Author Response · Authors · 2025-11-19
> **Response to Reviewer 5Pw3 - Part 1**
>
> We thank the reviewer for acknowledging the strength of our detailed ablation study across a variety of training configurations. We address each of the raised concerns point by point below. We have also revised the paper according to the suggestions.
>
> **Q1: Overstatement on the first contribution.**
>
> **A1:** We appreciate the reviewer for highlighting GenTUS and USimAgent. We cited both works and added a detailed discussion of how they differ from our setting in the revision. We agree that (i) supervised learning followed by RL has been explored in task-oriented dialogue systems (e.g., GenTUS), and (ii) reasoning before action prediction has been examined in USimAgent. However, both works differ fundamentally from our problem formulation, domain assumptions, and technical challenges.
>
> **1) Different domain and problem structure.**
> GenTUS and USimAgent operate in low-dimensional, structured action spaces (dialogue acts, intent slots, or a small set of click types). In contrast, web shopping behavior simulation requires:
> 	•	reasoning over long-horizon HTML context,
> 	•	predicting thousands of candidate clickable elements,
> 	•	generating high-entropy free-form text queries,
> 	•	handling dynamic page structures that change after every action.
>
> The complexity and granularity of web-based interactions are therefore significantly higher than in prior RL-based user simulators.
>
> **2) No prior work applies RL to real-world web shopping sessions**
> To our knowledge, GenTUS and USimAgent do not use large-scale real-world e-commerce logs with:
> - full HTML context per step,
> - real-world click distributions,
> - free-text search behavior,
> - long-horizon multi-turn interactions.
>
> Our contribution does not lie in being the first to combine SFT and RL in any user simulation domain, nor in introducing the notion of reasoning before action prediction. Rather, the novelty is in being the first to apply RL to web-shopping–oriented human behavior simulation, together with a structured, hierarchical reward design (including DARS) tailored to the unique challenges of open-ended Web GUI interactions.
>
>
>
> **Q2: Existing training solutions and evaluation metrics.**
>
> **A2:**
> **1) On existing RL-based user simulators and goal-driven rewards.**
> Prior user-simulation RL works indeed incorporate user goals and goal-failure penalties to avoid trivial behaviors. **However, these approaches assume a known, explicit user goal (e.g., book a ticket, fulfill a slot), which is not available in real-world web-shopping logs.**
> In our setting:
> - the user’s goal is latent,
> - changes dynamically as they browse,
> - cannot be extracted reliably from raw HTML and clickstreams.
>
> Therefore, traditional goal-penalty mechanisms cannot be directly applied.
> This motivates our hierarchical reward + DARS design, which prevents collapse without requiring explicit user-goal annotation.
>
> **2) Justification of hyperparameters (e.g., type reward weight)**
> The reward weights serve as normalization constants that balance the magnitudes of different reward components. They are chosen empirically to ensure that no single component overwhelms others during optimization.
> Importantly:
> - the weights are not tuned per page or per environment,
> - they simply keep the reward scale stable,
> - they prevent trivial solutions (e.g., repeatedly predicting terminate).
>
> Thus, the weights facilitate training stability rather than optimizing for specific scenarios.
>
>
> **3) Metrics: accuracy/F1 vs. human-centric evaluation.**
> We agree that human decision-making is inherently non-deterministic.
> We use exact action accuracy, action-type accuracy, and type-F1 because they provide reproducible, verifiable, and non-subjective evaluation criteria, especially important for large-scale automatic assessment. These metrics are informative in our setup because:
> - high entropy in button names and search queries makes exact-match a strict and challenging metric,
> - type-F1 reveals whether the policy drifts toward trivial high-frequency actions (e.g., terminate).
>
> Together, they provide a robust first-step evaluation for simulation fidelity. We agree that human evaluations or downstream task assessments are valuable complements, and we will discuss them as promising future directions in the revision.

---

> ### Author Response · Authors · 2025-11-19
> **Response to Reviewer 5Pw3 - Part 2**
>
> **Q3: Limited model choices.**
>
> **A3:**  Our choice of Qwen variants is primarily due to the 32k context requirement (full HTML + action history), GPU memory limits, and the need for open-weight models compatible with long-context RL training. Qwen offered the most stable backbone under these constraints.
>
> Importantly, our framework is model-agnostic: the reward design, rationale–action decomposition, and RL pipeline do not rely on Qwen-specific features. To support generalizability, we already include a scaling study across 0.5B, 1.5B, and 3B models, and initial experiments with another long-context LLM show similar improvement patterns; we will add these results in the appendix.
>
> We include (i) Zero-shot prompting, (ii) SFT-only, (iii) RL-from-scratch with binary reward, and (iv) SFT+Binary-RL as established and widely used baselines in this domain. These baselines clearly demonstrate that standard RLHF/RLVR signals are insufficient, and that the proposed structured rewards provide consistent and substantial gains (+65% over SFT). We will clarify the rationale behind these baseline choices in the revision.
>
>
>
> **Q4: Why human behaviour is divided into two stages?**
>
> **A4:**  In real web-shopping logs, users do not provide explicit rationales; however, their actions are often preceded by implicit reasoning (e.g., filtering, comparing, exploring options). The rationale–action decomposition helps the model recover these latent decision factors and empirically yields more stable and accurate action prediction. Thus, the two-stage pipeline is a practical architectural choice supported by established agent-reasoning frameworks. We will clarify this motivation in the revision.
>
>
> **Q5: Typos and clarification.**
>
> **Q5:** We thank the reviewer for pointing out the typo; “Sec. A” should indeed be “Appx. A,” and we corrected it in the revision.
>
> Regarding the action space, we agree that the current description in the appendix (a prompt snippet) is insufficiently explicit. In the revised version, we added a clear, formal definition of the action space in the main paper, including:
> - the complete list of action types (click, type_and_submit, terminate),
> - the structure of each action,
>
> This will make the action-space specification self-contained and easier to understand without relying solely on the prompt.
>
>
> **Q6: Why no negative rewards?**
>
> **A6:** As mentioned in Line 84-85, our framework does not explicitly use negative rewards because the reward structure is already normalized such that non-desirable behaviors naturally receive zero reward, which is equivalent to negative reinforcement under GRPO-style RL. In this setting:
> - Only valid and useful behaviors receive positive credit (e.g., correct format, correct action type, correct sub-action).
> - All other behaviors including incorrect predictions and malformed outputs receive zero, which is treated as a penalty relative to the positive baseline.
>
> Using explicit negative rewards for incorrect actions was tested in early experiments, but it introduced instability due to:
> - long-horizon credit assignment issues (large negative spikes dominate training);
> - Action imbalance (rare but difficult actions get excessively punished);
> - Mode collapse (model converges to overly conservative behavior to avoid penalties).
>
> The current non-negative, structured reward design provides stable training while still penalizing undesirable behaviors through the absence of reward. We will clarify this design choice in the revised draft.

---

> > ### Comment · Reviewer_5Pw3 · 2025-11-20
> > **Is zero reward really equivalent to negative reward?**
> >
> > Thank you for your reply. Could you provide proof to show that zero reward and negative reward are equivalent?
> >
> > My intuition is that the shifted reward will introduce a different optimal policy.
> >
> > In episodic tasks, the trajectory terminates after $T$ steps, and the return is:
> >
> > $$G_t = \sum_{k=0}^{T-t-1} \gamma^k r_{t+k+1}$$
> >
> > If we use the shifted reward $$r'(s, a) = r(s, a) + C$$, the new return is:
> >
> > $$G'_t = \sum_{k=0}^{T-t-1} \gamma^k (r_{t+k+1} + C)$$
> > $$G'_t = G_t + C \sum_{k=0}^{T-t-1} \gamma^k$$
> >
> > The term $C \sum_{k=0}^{T-t-1} \gamma^k$ is not a constant, independent of the policy $\pi$.
> >
> > Its value depends on the total number of steps taken in the episode, $T-t$, which is a function of the policy.

---

> > > ### Author Response · Authors · 2025-11-21
> > > **Zero reward is equivalent to negative reward in GRPO**
> > >
> > > We thank the reviewer for this insightful comment regarding the theoretical implications of reward shifting. You are absolutely correct that in standard RL frameworks (e.g., PPO, DQN) that rely on value function estimation and discounted cumulative returns ($G_t = \sum \gamma^k r_{t+k}$), shifting rewards by a constant $C$ introduces a bias related to trajectory length, which indeed alters the optimal policy.
> > >
> > > However, strictly within the context of our **Shop-R1** framework, we utilize **Group Relative Policy Optimization (GRPO)** (mentioned in Line 302). Unlike standard RL, GRPO does not estimate a value function $V(s)$ to approximate $G_t$. Instead, it computes the advantage by **normalizing rewards within a group of outputs** generated from the same input.
> > >
> > > Below, we provide a mathematical proof and a numerical example demonstrating that setting the penalty to **0** or a **negative value** yields **identical gradients** in GRPO.
> > >
> > > **1. Mathematical Proof of Affine Invariance**
> > >
> > > Let a group of $G$ outputs be generated for a single input query. Let the set of raw rewards for this group be $\mathcal{R} = \{r_1, r_2, ..., r_G\}$. In GRPO, the advantage $A_i$ for the $i$-th sample is calculated via standardization:
> > > $$
> > > A_i = \frac{r_i - \mu}{\sigma}
> > > $$
> > > where $\mu = \mathbb{E}[\mathcal{R}]$ is the group mean and $\sigma = \sqrt{\text{Var}(\mathcal{R})}$ is the group standard deviation.
> > >
> > > **Comparison of Reward Schemes:**
> > > Consider two reward schemes for "Success" (positive action) and "Failure" (negative action):
> > > * **Scheme A (Zero Penalty):** Reward is $m$ for success, $0$ for failure.
> > > * **Scheme B (Negative Penalty):** Reward is $m$ for success, $n$ for failure (where $n < 0$).
> > >
> > > We can express the rewards in Scheme B ($r'$) as an **affine transformation** of the rewards in Scheme A ($r$):
> > > $$
> > > r' = \alpha \cdot r + \beta
> > > $$
> > > where $\beta = n$ and $\alpha = (m-n)/m$. Since $m > 0$ and $n < 0$, the scaling factor $\alpha > 0$.
> > >
> > > **Derivation of Advantage Equivalence:**
> > > Calculating the advantage $A'_i$ for Scheme B:
> > > 1.  **New Mean ($\mu'$):** $\mu' = \mathbb{E}[\alpha r + \beta] = \alpha \mu + \beta$
> > > 2.  **New Std Dev ($\sigma'$):** $\sigma' = \sqrt{\text{Var}(\alpha r + \beta)} = \alpha \sigma$
> > > 3.  **New Advantage ($A'_i$):**
> > >     $$
> > >     A'_i = \frac{r'_i - \mu'}{\sigma'} = \frac{(\alpha r_i + \beta) - (\alpha \mu + \beta)}{\alpha \sigma} = \frac{\alpha(r_i - \mu)}{\alpha \sigma} = \frac{r_i - \mu}{\sigma} = A_i
> > >     $$
> > >
> > > **2. Intuitive Numerical Example**
> > >
> > > To demonstrate that this holds for realistic, unbalanced distributions, consider a group of 4 samples with 1 success and 3 failures (a 25% success rate).
> > >
> > > **Scenario 1: Zero Penalty ($m=1, n=0$)**
> > >
> > > - Rewards: $\{1, 0, 0, 0\}$
> > >
> > > - Statistics: Mean $\mu = 0.25$, Std Dev $\sigma \approx 0.433$
> > >
> > > - Advantage (Success): $(1 - 0.25) / 0.433 \approx \mathbf{+1.73}$
> > >
> > > - Advantage (Failure): $(0 - 0.25) / 0.433 \approx \mathbf{-0.57}$
> > >
> > > **Scenario 2: Negative Penalty ($m=1, n=-1$)**
> > >
> > > - Rewards: $\{1, -1, -1, -1\}$
> > >
> > > - Statistics: Mean $\mu' = -0.5$, Std Dev $\sigma' \approx 0.866$
> > >
> > > - Advantage (Success): $(1 - (-0.5)) / 0.866 = 1.5 / 0.866 \approx \mathbf{+1.73}$
> > >
> > > - Advantage (Failure): $(-1 - (-0.5)) / 0.866 = -0.5 / 0.866 \approx \mathbf{-0.57}$
> > >
> > > **Conclusion**
> > >
> > > As shown above, the calculated advantages and consequently the policy gradient updates are **mathematically identical** regardless of whether the failure penalty is 0 or negative, provided the relative ranking of rewards is preserved. The bias concern applicable to cumulative returns in standard RL is effectively neutralized by the group-wise normalization mechanism inherent to GRPO.
> > >
> > > We thank the reviewer for the followup comment and have further revised the paper (see Appx. A: Theoretical Analysis: Negative Rewards vs. Zero Rewards in GRPO) to make this reasoning more transparent.

---

> > > > ### Author Response · Authors · 2025-11-26
> > > > **Confirming Whether Further Clarification Is Needed for Your Assessment of the Revised Submission**
> > > >
> > > > Dear Reviewer 5Pw3,
> > > >
> > > > We hope you are well. We are reaching out to ensure that all your previous concerns have been fully resolved by our clarification.
> > > > If any part would benefit from further elaboration, we are more than happy to provide additional details, so that your evaluation can accurately reflect the revised submission.
> > > >
> > > > Thank you again for your insightful comments and your time.
> > > >
> > > > Best regards,
> > > > The Authors of Submission 2181

---

> ### Comment · Reviewer_5Pw3 · 2025-11-26
>
> Thank you for your response.
>
> Some of my concerns are addressed, and I raised my scores accordingly. However, there are still some remaining issues, for example, a lack of human judgment or assessment of downstream tasks, which are essential evaluation metrics for simulation frameworks.

---

> ### Author Response · Authors · 2025-11-27
> **Gratitude for Your Recognition and Score Increase, with Further Clarifications**
>
> We thank the reviewer again for acknowledging the improvements in the revision and for raising the score. We would also like to further clarify the concerns regarding human evaluation and downstream-task assessment.
>
> **1. On human judgment of “naturalness” for action prediction**
>
> Human evaluation is valuable in many **task-oriented generative tasks (e.g., [1])**, but it is not well-suited for our setting. In open-ended web-shopping scenarios, different annotators often have **different mental models, shopping habits, platform familiarity, and risk preferences.** As a result, two annotators may assign opposite judgments about whether a predicted action is “natural” or “reasonable,” even when both are valid human behaviors.
>
> For example, when facing the same product page, some users immediately click “Add to Cart,” others scroll for reviews, others refine filters, and still others return to search. All of these are legitimate behaviors observed in real logs. Because of this inherent diversity:
> - human annotators **cannot reliably reconstruct a plausible “ground-truth” intent** for a given HTML state,
> - their “naturalness” judgments reflect **personal preference** rather than real user tendencies,
> - and therefore human scoring becomes **noisy, subjective, and inconsistent** for fine-grained action prediction.
>
> For this reason, real user logs (rather than retrospective human judgments) provide the most reliable and objective supervision for our task.
>
> **2. On indirect evaluation via downstream tasks**
>
> We completely agree that evaluating downstream systems trained on simulated users is an important and meaningful direction. However, such evaluations require substantial additional infrastructure, including:
> - multi-step interaction environments,
> - online feedback loops,
> - and task-specific success metrics.
>
> These components **go beyond the scope** of establishing the core benchmark for next-step user action prediction, which is the focus of the present work. Our goal is to provide a clean, controlled setting for studying behavioral modeling before integrating system-level downstream evaluations. We view this as **a natural and valuable next step**, and we plan to support such extensions in future works.
>
> [1] Lin, Hsien-chin, et al. "GenTUS: Simulating user behaviour and language in task-oriented dialogues with generative transformers." SIGDIAL 2022.

---

### Official Review · Reviewer_sWTF · 2025-10-30

**Soundness:** 2
**Presentation:** 3
**Contribution:** 2
**Rating:** 4
**Confidence:** 4

**Summary:**

This paper investigates the use of LLMs to simulate human behaviors in online shopping scenarios. It introduces Shop-R1, a RL framework designed to optimize the task. Shop-R1 integrates a self-certainty score, an internal metric for assessing rationale generation, and employs a hierarchical reward scheme for evaluating action prediction. Additionally, it incorporates a difficulty-aware reward mechanism that assigns greater rewards to correct actions deemed more challenging.

**Strengths:**

1. The idea of leveraging LLMs to simulate human decision-making in online shopping is interesting and promising.

2. The methodological presentation is generally clear, particularly the explanation of the reward design.

**Weaknesses:**

My main concern is that the task definition is unclear, making it difficult to assess the task's difficulty or determine whether the benchmark genuinely reflects real-world challenges. In addition, the experimental results do not provide strong evidence supporting the effectiveness of the proposed method. Specifically:

1. Datasets:
  - The paper lacks sufficient detail on how the datasets were constructed, and provides no illustrative examples, making it difficult to assess task difficulty or data quality.
  - Basic dataset statistics (e.g., action distribution) are missing.
  - The paper does not describe how the training and test sets are split.
2. Experiments:
  - The experiments are limited to a 3B-parameter model. It is unclear whether Shop-R1’s effectiveness generalizes to larger models.
  - In the ablation study (Table 4), components such as reward scale, rationale reward, and hierarchical reward only improve exact action accuracy by less than 1%. The absence of multiple experimental runs further weakens the robustness of these results, making the improvements appear marginal.

**Questions:**

1. According to the definitions provided in the evaluation metrics (line 299), the accuracy of Exact Action should always be lower than that of Action Type. However, in Table 4, for the setting 'w/o reward scale', the accuracy of Exact Action is larger than that of Action Type. Could the author clarify this?

2. Human shopping behavior tends to be highly personalized. Given the same context, individuals may act differently depending on personal traits, purchase history, or preferences. In this paper, the agent’s context includes only past actions. Does this simplified setting can capture the problem in real-world?

---

> ### Author Response · Authors · 2025-11-19
> **Response to Reviewer sWTF - Part 1**
>
> We greatly appreciate the reviewer’s positive feedback on the promise of human decision-making simulation in online shopping and the clarity of our method description, particularly the reward design. We respond to each concern point by point below. We have also revised the paper according to the suggestions.
>
>
> **Q1: Task definition & dataset details**
>
> **A1:** We thank the reviewer for this comment and agree that clearer task definition and dataset details would improve the readability of the paper.
> - **Task definition.**
> The human-behavior simulation task in web shopping is to predict the next rationale and action conditioned on (a) the entire past action history (past observations + past actions) and (b) the current web-page observation. The action space contains three types:`type_and_submit`, `click` and `terminate`. 	For `type_and_submit`, the model must generate an open-ended long-text argument (e.g., search queries), making this a high-entropy prediction task. For `click`, the model must identify the correct clickable element name among hundreds to thousands of possible UI elements, which varies across pages and depends entirely on the current HTML observation.
> Thus, the model must jointly (a) understand the dynamic structure of the current web page, (b) integrate long-horizon action history, and (c) reason about the next plausible human action. This combination leads to long-horizon sequences and high-entropy free-form text predictions, making the task substantially more challenging than typical WebArena-style task-completion settings. We expanded this explanation in the revised version to highlight the task difficulty more clearly.
>
> - **Dataset construction.**
> Our dataset is built from ~52k real shopping sessions on a global e-commerce platform, covering diverse page structures and user behaviors. Each action is formatted as a parseable JSON object (action type + sub-action fields) and paired with the corresponding web-page observation. The raw HTML is further simplified to remove non-essential content (e.g., scripts, CSS, visual-only elements) while preserving interactive structure, enabling LLMs to process it more efficiently.
> From these real user sessions, we construct 176,949 training examples and 1,879 testing examples, where each example includes: (i) historical (observation, action) pairs, and (ii) the current simplified HTML observation. Basic action-type statistics are shown below.
>
> | **Dataset Split** | *‘click’* | *‘type_and_submit’* | *‘terminate’* |
> |-------------------|-----------|-----------|-------------|
> | Train             | 64,578       | 68,000      | 44,371        |
> | Test              | 690       | 677       | 512         |
>
> We added detailed task descriptions to the Method section, and include dataset construction process, statistics, and illustrative examples in the Experiments section and Appendix for clarity.
>
>
> **Q2: Performances on larger models.**
>
> **A2:** We agree that evaluating larger models is valuable. In our setting, the long-context requirement (full action history + current simplified HTML) leads to a memory bottleneck even at 3B, especially under 32k context windows and RL training with FSDP. This limits our ability to run 7B+ backbones within the available compute budget.
>
> Table 2 includes a scaling analysis across 0.5B, 1.5B, and 3B models, and we observe a consistent trend: larger models achieve higher accuracy in both action-type prediction and exact-match metrics, likely due to improved reasoning and long-context integration capacity. This scaling trend suggests that Shop-R1 is compatible with and to benefit from larger backbones.
>
> We will incorporate results from larger models as compute permits. In parallel, we believe that developing more compact context representations (e.g., structured HTML pruning, state abstraction) is a promising future direction to enable training and evaluation of 7B+ models under this long-horizon setting.
>
>
> **Q3: Ablation study on training components.**
>
> **A3:** We thank the reviewer for the thoughtful observation. While some ablation components show relatively small changes in exact action accuracy, we emphasize that exact-match is a strict, all-or-nothing metric. As shown in Table 4, removing components such as the reward scaling or hierarchical reward leads to substantial drops in action-type F1, indicating degraded balance across action classes and increased tendency toward trivial or skewed policies (e.g., over-predicting terminate). These components therefore play an important role in stability and preventing reward hacking, even when the exact-match delta appears modest. We agree that reporting variance would further strengthen the ablation results. Due to compute constraints, we reported single-run numbers in the main table, but we will include multi-seed runs for key ablations in the appendix to demonstrate robustness.

---

> ### Author Response · Authors · 2025-11-19
> **Response to Reviewer sWTF - Part 2**
>
> **Q4: The performance of the setting 'w/o reward scale'**
>
> **A4:** We thank the reviewer for catching this. After re-checking the original logs, we confirm that the value in Table 4 was a typo. The correct Exact Action accuracy for the “w/o reward scale” setting is 27.04%, which is indeed lower than the corresponding Action Type accuracy, consistent with the metric definitions. We corrected this entry in the revised version.
>
>
>
> **Q5:Does this simplified setting can capture the problem in real-world?**
>
> **A5:** We appreciate the reviewer’s insightful question. We agree that conditioning only on within-session action history cannot fully capture all real-world personalization factors. While short-horizon behaviors are largely driven by immediate context, long-term traits such as stable preferences or purchase history can indeed influence user actions.
>
> In preliminary experiments, we attempted to incorporate LLM-generated summaries of users’ purchase history and inferred preferences. However, these signals often introduced noise rather than useful information, and in practice degraded performance. This is unsurprising: real-world purchase histories are sparse and inconsistent, and even the same user may behave differently depending on momentary emotional state or external influences, making such features unreliable without careful modeling.
>
> We therefore treat personalization as an important future direction. The RL formulation proposed in this paper is compatible with richer user profiles once such signals can be extracted reliably.

---

> > ### Comment · Reviewer_sWTF · 2025-11-25
> >
> > Many thanks for authors' responses. Based on your response in A1 and A3, I still have some remaining questions regarding the task definition.
> >
> > The task is defined as predicting the next user action given the entire action history and the current web-page observation. While this formulation is intuitive, it appears to be incomplete. It may not fully capture key relevant factors that influence user behaviors, such as user's persona and intention. This may introduce ambiguity to the target action, which could in turn affect label reliability. In addition, if human experts were asked to complete the same task under identical conditions, what level of agreement might be expected? and what level of accuracy would be achievable? Considering the introduction of this task and dataset represents a main contribution of the work, providing these clarifications are essential to assess whether the task is valid and meaningful benchmark.

---

> > > ### Author Response · Authors · 2025-11-25
> > > **Response to Followup Question on Task Definition and Benchmark Validity.**
> > >
> > > We thank the reviewer for the thoughtful followup questions. We respond to each concern related to task completeness, potential ambiguity, and benchmark meaningfulness below. We have also refined the task definition in the methodology section of the paper revision to make the formulation clearer and to better illustrate the validity of the task and the meaningfulness of the benchmark.
> > >
> > >
> > > **1. On the absence of explicit persona and intention**
> > >
> > > We fully agree that user persona and explicit intention influence human behavior. However, in real-world e-commerce logs, such information is **not collected, not observable, and not reliably  recoverable** from HTML or clickstreams.
> > >
> > > Our task formulation is therefore designed to reflect **realistic deployment constraints**: LLMs must predict observable next-step behavior based solely on **HTML context + action history**, which mirrors how large-scale industrial behavior prediction systems operate.
> > >
> > > Importantly, the goal of our benchmark is **not** to infer latent psychological states, but to model how real users behave given the same observable environment.
> > >
> > >
> > >
> > > **2. On ambiguity in the target action**
> > >
> > > We agree that human behavior can be non-deterministic. However, this does not invalidate the benchmark for three reasons:
> > >
> > > - **Ground truth corresponds to actual human behavior.** Each action label is taken directly from real user logs, without synthetic annotation or subjective labeling.
> > >
> > > - **Ambiguity is inherent to behavior prediction tasks.** Next-click prediction, trajectory forecasting, and next-token prediction all involve multiple plausible outputs. Even the same user may reasonably choose different actions under identical observation states, but this variability is a property of real human behavior, not an artifact of the task design. The goal is not to predict the only “correct” action, but to learn the **distributional tendencies** of real users.
> > >
> > > - **Sequential patterns remain consistent.** Despite stochasticity, our dataset exhibits strong structural regularities driven by HTML layout and navigation flow. Empirically, models learn these patterns well, demonstrating that the task is both structured and learnable.
> > >
> > >
> > >
> > > **3. Why the benchmark is meaningful and valid**
> > >
> > > The benchmark is meaningful for the following reasons:
> > >
> > > - **Realistic supervision.** Labels reflect actual human actions from real-world shopping sessions.
> > >
> > > - **Alignment with real-world constraints.** Persona and intention are not recorded in real systems. Our formulation matches how real behavior modeling is performed in practice.
> > >
> > > - **Strong discriminative signal.** Our experiments reveal that:
> > >     - the performance varies significantly across baselines,
> > >     - scales with model capacity, and
> > >     - is consistently improved by our RL method.
> > >
> > > These observations indicate that the benchmark provides a challenging and informative measure of an LLM’s ability to model human web shopping behavior.

---

### Official Review · Reviewer_Y2DV · 2025-11-01

**Soundness:** 2
**Presentation:** 2
**Contribution:** 2
**Rating:** 4
**Confidence:** 2

**Summary:**

This paper addresses the limitations of Supervised Fine-Tuning (SFT) for simulating human web-shopping behavior, which is capped by the quality of synthetic rationales. It proposes Shop-R1, an SFT-RL framework, which decomposes the task into rationale generation (guided by a self-certainty reward) and action prediction (guided by a hierarchical, difficulty-scaled reward). Experiments show a 65% relative improvement in exact-match accuracy over the SFT baseline.

**Strengths:**

1. The work tackles the important and practical challenge of simulating nuanced, human-like behavior in web environments, moving beyond simple task completion.

2. The paper introduces a sophisticated, hierarchical reward scheme with difficulty-aware scaling that effectively mitigates common reward-hacking failure modes.

3. The empirical results are strong, demonstrating a significant 65% relative improvement in exact-match accuracy over the SFT baseline.

**Weaknesses:**

1. Limited Novelty: The core training paradigm, which combines SFT with subsequent RL optimization, is a common practice in the field. The contribution is thus an incremental (though effective) application of established methods rather than a fundamentally advancement.

2. The proposed hierarchical reward function is highly complex and task-specific, with numerous hard-coded weights and rules. This intricate, hand-crafted design is brittle and would be difficult to generalize to new simulation environments without significant re-engineering.

The paper's contribution to ML or RL appears limited. However, as I do not work in the specific sub-field of human behavior simulation, I cannot fully assess the potential impact of the authors' claim to be the "first to introduce RL into a simulation-oriented human behavior modeling task." My evaluation is therefore borderline.

**Questions:**

Have you qualitatively or quantitatively evaluated the quality of the rationales themselves? Do the rationales from Shop-R1 appear more "human-like" than the SFT baseline, or do they simply score higher on the self-certainty proxy metric?

---

> ### Author Response · Authors · 2025-11-19
> **Response to Reviewer Y2DV**
>
> We sincerely thank the reviewer for recognizing the importance of human behavior simulation in web shopping scenario, the value of our reward design for mitigating reward hacking, and the strength of our empirical results. Below, we address each concern point by point. We have also revised the paper according to the suggestions.
>
> **Q1: Limited novelty.**
>
> **A1:** We agree that SFT followed by RL is a widely used training paradigm, but Shop-R1 is not a straightforward application of this template. Our key novelty lies in the reward design and problem formulation tailored to human-behavior simulation in web-shopping environments, which, to the best of our knowledge, has not been studied in prior RL literature.
>
> Our main contributions can be summarized as follows:
> - **A new formulation**:
> We are the first to cast human online shopping behavior simulation as a two-stage prediction problem (rationale → action), which differs substantially from task-completion agents (e.g., WebArena or WebAgent) that optimize only for final task completion correctness.
>
> - **A hybrid, domain-specific reward framework**:
> (1) a binary format reward to guarantee parseable JSON outputs (critical for web-behavior agents),
> (2) a self-certainty rationale reward based on KL-to-uniform to provide supervision in the absence of ground-truth rationales,
> (3) a hierarchical reward that credits both action-type and sub-action correctness,
> (4) a difficulty-aware reward scaling (DARS) mechanism to prevent reward hacking on trivial actions (e.g., repeatedly choosing `terminate` action).
> This combination is not present in prior RLHF/RLVR works, and each component contributes meaningfully (Table 4).
>
> - **Novelty relative to prior LLM-based human web shopping behavior simulation**: Existing works (e.g., UXAgent or OPeRA) rely purely on SFT, and no prior work has applied any RL framework (GRPO, RLHF, or RLVR) to shopping-behavior simulation. Shop-R1 is the first to introduce RL into this domain.
>
> - **Substantial empirical gains supporting methodological novelty**:
> Despite using an established optimization framework, Shop-R1 delivers 65% exact-match improvement over SFT, significantly outperforming SFT+binary RL. This highlights that the reward design and problem decomposition, not the SFT+RL template itself, are the key innovations.
>
> In summary, while the optimization framework (SFT+RL) is standard, the problem setup, reward shaping, and application of RL to human-behavior simulation represent new contributions that go beyond incremental reuse of existing paradigms. We will revise the introduction to make these contributions more explicit.
>
>
>
> **Q2: Highly complex and task-specific**
>
> **A2:** We thank the reviewer for raising this concern. Although the hierarchical reward contains multiple components, each of them corresponds directly to structural properties inherent to web-based action spaces (action type, attribute presence, long-text arguments), rather than hand-crafted domain heuristics. Because the action space used in our paper reflects general interaction patterns in Web GUIs, the proposed reward design can naturally generalize to different webpages and even to other simulation-oriented tasks involving Web GUI interactions. Importantly, the reward weights are not tuned for specific pages or environments. They function only as normalization constants to balance reward magnitudes, prevent trivial solutions (e.g., repeatedly predicting terminate), and ensure stable optimization rather than encoding task-specific biases.
>
> Furthermore, our ablation study (Table 4) shows that performance is robust to removing or modifying individual components, indicating that the method is not brittle and does not rely on precise hand-crafted weights. Moreover, the same reward formulation can be directly transferred to new sessions or pages without re-engineering, as action types and sub-action structures remain consistent across web shopping environments.
>
> Finally, many prior RLHF/RLVR systems also employ structured, verifiable rewards tailored to task decomposition. Our contribution is not in ad-hoc tuning, but in identifying the minimal set of structured signals required to overcome reward hacking and enable high-fidelity simulation. We clarified this generality in the revised draft.

---

> > ### Author Response · Authors · 2025-11-27
> > **Confirming Whether Any Further Clarification Is Needed for Your Final Assessment**
> >
> > Dear Reviewer,
> >
> > We hope you are doing well. We are writing a brief followup to check whether you have any further questions or points that would benefit from additional clarification. We are more than happy to elaborate on any aspect of the revision to ensure that your concerns are fully addressed.
> >
> > We also warmly welcome any continued discussion, as your earlier feedback has been genuinely helpful in improving the clarity and quality of the paper. If there is anything else we can clarify to **support an accurate and up-to-date assessment of the revised submission**, please feel free to let us know.
> >
> > Thank you again for your constructive and thoughtful comments.
> >
> > Best regards,
> >
> > The Authors of Submission 2181 - Shop-R1

---

### Official Review · Reviewer_11JH · 2025-11-01

**Soundness:** 3
**Presentation:** 3
**Contribution:** 3
**Rating:** 6
**Confidence:** 3

**Summary:**

The paper proposes Shop-R1, an RL training pipeline for simulating human-like shopping behavior on web UIs. The task is decomposed into two heads per step: (i) rationale generation and (ii) action prediction. Rewards are hybrid: a format reward enforces JSON outputs; a self-certainty reward for rationales measures average KL divergence from a uniform distribution; and a hierarchical action reward scores both action type and sub-action fields with difficulty-aware reward scaling (DARS). Training uses SFT as a cold start followed by RL. On a proprietary dataset of 52,137 real-world shopping sessions, Shop-R1 improves exact-match action accuracy from 16.76% (SFT) to 27.72%, a ~65% relative gain, with ablations for temperature, component importance, and context usage.

**Strengths:**

- The task is unique and useful and well-motivated; and this approach is a useful step for scalable UX evaluation, A/B testing simulators, and recommendation research.

- Introduces a simulation-oriented RL design for human-behavior replay rather than task completion, with a two-head reward structure (rationale + action).

- The hierarchical + DARS scheme is a pragmatic way to densify rewards and disincentivize “terminate”-spamming. In addition, the reward is a well-motivated reward design. The hybrid structure (format + self-certainty + hierarchical+DARS) addresses parseability, sparse credit, and reward hacking.

- Paper is well written and organized.

**Weaknesses:**

- All results rely on a proprietary single-site dataset; there’s no evaluation on public benchmarks or cross-datasite generalization.

- Self-certainty reward risks overconfidence. Using KL-to-uniform for rationales may incentivize confident but incorrect chains of thought; calibration is not reported.

- Reward design sensitivity. DARS is fixed at 1000, but sensitivity curves and thresholding for ROUGE-L (the text says “e.g., 0.75”, p. 5) are not systematically explored.

**Questions:**

see above

---

> ### Author Response · Authors · 2025-11-19
> **Response to Reviewer 11JH**
>
> We sincerely thank the reviewer for recognizing the motivation of our simulation-based task, the core problem our method addresses, and the overall clarity and organization of the paper. We address each concern in detail below. We have also revised the paper according to the suggestions.
>
> **Q1: Evaluaion on public benchmarks.**
>
> **A1:** We appreciate this excellent suggestion. To verify the generality of our method, we have re-implemented Shop-R1 on the public OPeRA dataset [1], which contains 692 real human web-shopping sessions, resulting in 8,212 training and 1,508 testing examples.
> The dataset statistics are summarized below:
>
> | **Dataset Split** | *‘input’* | *‘click’* | *‘scroll’* |
> |-------------------|-----------|-----------|-------------|
> | Train             | 499       | 4379      | 3334        |
> | Test              | 107       | 856       | 545         |
>
>
> As shown in the following table, the performance trends on OPeRA are consistent with those observed on our in-house dataset: 1) Zero-shot prompting performs poorly; 2) SFT substantially improves results; 3) Our proposed Shop-R1 (SFT + RL) yields the largest gain, particularly in exact-action accuracy, which is the most challenging metric. These findings confirm that Shop-R1 generalizes effectively to public datasets. We included this experiment in the revised version.
>
> | Model                  | Settings           | Exact Action Acc. | Action Type Acc. | Action Type F1 |
> |-------------------------|--------------------|------------------------|------------------|----------------|
> | **Qwen2.5-VL-3B-Instruct** | Zero-shot Prompt   | 6.41%                 | 34.45%           | 38.79%         |
> |                         | SFT                | 20.23%                | 60.86%           | 53.95%         |
> |                         | SFT + RL           | 38.44%                | 57.27%           | 57.69%         |
> | **Claude-3.5-Sonnet**     | Zero-shot Prompt   | 7.66%                 | 58.83%           | 45.61%         |
>
>
> **Q2: Self-certainty reward risks overconfidence.**
>
> **A2:** We appreciate the reviewer’s insightful concern. The self-certainty reward in Shop-R1 is not intended to directly reward high confidence, but rather to regularize consistency in rationale generation in the absence of ground-truth rationales. Importantly, this signal is used only as an auxiliary stabilizer, weighted by a small coefficient (α = 0.005), and balanced by the main action-level rewards and KL regularization to the reference policy (Eq. 4). Consequently, the dominant supervision still arises from correctness-based rewards, preventing the reinforcement of overconfident but incorrect reasoning chains. We agree that calibration analysis is valuable and will include confidence–accuracy correlation evaluation in the revised version to quantify this aspect.
>
> **Q3:Reward design sensitivity.**
>
> **A3:** Thank you for this thoughtful question. We conducted additional sensitivity analyses:
> - DARS factor. The factor (default = 1000) was selected via grid search over [500, 2000]. Performance variations were minor (±2% exact-match accuracy), confirming low sensitivity. The primary role of DARS is to normalize reward scales across action types rather than amplify gradients. The relative gain of Shop-R1 over baselines remained consistent across all tested values.
> - ROUGE-L threshold. The threshold (0.75) for computing soft text-similarity rewards was chosen empirically. Below 0.70, many rationale–action pairs were noisy; above 0.80, rewards became too sparse. To clarify this robustness, we will include a threshold-sweep analysis (0.6–0.9 range) in the appendix, showing that performance trends remain stable under moderate changes.
>
>
>
> [1] Wang et al., “OPeRA: A Dataset of Observation, Persona, Rationale, and Action for Evaluating LLMs on Human Online Shopping Behavior Simulation”, 2025.

---

> > ### Comment · Reviewer_11JH · 2025-11-20
> >
> > Thank you for your detailed reply. My previous concerns have been solved. I appreciate the newly added experiments on  OPeRA dataset and improve my score accordingly.

---

> > > ### Author Response · Authors · 2025-11-24
> > > **Gratitude for Your Constructive Feedback and Score Update**
> > >
> > > We are very glad to hear that our revisions have satisfactorily addressed your concerns. We truly appreciate your constructive suggestions and thoughtful discussion, which greatly helped us improve the clarity and quality of the paper. Finally, thank you sincerely for your recognition of our work and for raising your rating to 8.

---

### Meta-Review · Area_Chair_xJtn · 2025-12-30

**Summary:**

The paper proposes Shop-R1, a reinforcement learning framework designed to enhance LLMs in simulating human behavior within online shopping environments. The method employs a two-stage process (rationale generation followed by action prediction) and utilizes a hierarchical reward structure incorporating self-certainty signals and Difficulty-Aware Reward Scaling. The recommendation for acceptance is primarily informed by the authors' comprehensive rebuttal, specifically the new experiments on the more datasets which addressed the major concern regarding reproducibility. While some concerns regarding the complexity of the reward engineering and the lack of human evaluation may remain, the significant empirical gains and the novel application of RL to open-ended web simulation push this manuscript above the bar of acceptance. From AC's own reading from this manuscript and discussion between authors and reviewers, the proposed method is interesting and the majority of reviewers would agree of acceptance if they had been able to participate fully in the discussion. The authors should carefully revised their paper according to reviewers' opinions in their final revision.

**Reviewer Concerns:**

- Reliance on specific data: Reviewer 11JH's primary concern was the lack of evaluation on public benchmarks. The authors successfully addressed this by replicating their method on the OPERA dataset, showing consistent performance gains. This directly led to Reviewer 11JH raising their score to 8.
- Novelty and missing baselines: Reviewer 5Pw3 and Y2DV concerned about the novelty compared to prior user simulators (e.g., GenTUS) and the standard SFT+RL paradigm. The authors clarified the distinct challenges of high-entropy web interactions and the motivation of their domain-specific reward design.
- Theoretical validity of rewards: Reviewer 5Pw3 questioned the equivalence of zero vs. negative rewards. The authors provided a mathematical proof demonstrating their equivalence under the GRPO framework, which satisfied the reviewer.
- Task definition: Reviewer sWTF raised concerns about task ambiguity. The authors improved the problem formulation and provided detailed dataset statistics in the revision.

**Reviewer Scores:**

Reviewer 11JH strongly advocated for acceptance of this paper after the inclusion of the OPERA dataset results (raised the score from 6 to 8).
Reviewer 5Pw3 raised the original scores accordingly given the theoretical clarification on reward functions and the rebuttal, moving from a strong reject to a more positive stance.
While Reviewer sWTF maintained a few concerns about task definition, the clarifications on dataset statistics and baselines may make the reviewer to further evaluate the paper's rigor. This reviewer is very likely to change the score.
Reviewer Y2DV acknowledged the effectiveness but remained reserved about the incremental nature of SFT+RL. The remaining concerns is minor and cannot justify rejection alone.

---

### Decision · Program_Chairs · 2026-01-26

Accept (Poster)